# Effect of Soil Drought Stress on Selected Biochemical Parameters and Yield of Oat × Maize Addition (OMA) Lines

**DOI:** 10.3390/ijms241813905

**Published:** 2023-09-09

**Authors:** Tomasz Warzecha, Jan Bocianowski, Marzena Warchoł, Roman Bathelt, Agnieszka Sutkowska, Edyta Skrzypek

**Affiliations:** 1Department of Plant Breeding, Physiology and Seed Science, University of Agriculture in Krakow, Łobzowska 24, 31-140 Kraków, Poland; tomasz.warzecha@urk.edu.pl (T.W.); roman.bathelt@student.urk.edu.pl (R.B.); agnieszka.sutkowska@urk.edu.pl (A.S.); 2Department of Mathematical and Statistical Methods, Poznań University of Life Sciences, Wojska Polskiego 28, 60-637 Poznań, Poland; 3Department of Biotechnology, The Franciszek Górski Institute of Plant Physiology, Polish Academy of Sciences, Niezapominajek 21, 30-239 Kraków, Poland; m.warchol@ifr-pan.edu.pl (M.W.); e.skrzypek@ifr-pan.edu.pl (E.S.)

**Keywords:** agronomic traits, drought stress, Grande I, maize, oat, OMA, phenolic compounds, soluble sugars

## Abstract

Plant growth and the process of yield formation in crops are moderated by surrounding conditions, as well as the interaction of the genetic background of plants and the environment. In the last two decades, significant climatic changes have been observed, generating unfavorable and harmful impacts on plant development. Drought stress can be considered one of the most dangerous environmental factors affecting the life cycle of plants, reducing biomass production and, finally, the yield. Plants can respond to water deficit in a wide range, which depends on the species, genetic variability within the species, the plant’s ontogenesis stage, the intensity of the stress, and other potential stress factors. In plants, it is possible to observe hybrids between different taxa that certain traits adopted to tolerate stress conditions better than the parent plants. Oat × maize addition (OMA) plants are good examples of hybrids generated via wide crossing. They can exhibit morphological, physiological, and biochemical variations implemented by the occurrence of extra chromosomes of maize, as well as the interaction of maize and oat chromatin. The initial goal of the study was to identify OMA lines among plants produced by wide crossing with maize. The main goal was to investigate differences in OMA lines according to the Excised Leaf Water Loss (ELWL) test and to identify specific biochemical changes and agronomic traits under optimal water conditions and soil drought. Additionally, detection of any potential alterations that are stable in F2 and F3 generations. The aforementioned outcomes were the basis for the selection of OMA lines that tolerate growth in an environment with limited water availability. The molecular analysis indicated 12.5% OMA lines among all tested descendants of wide oat-maize crossing. The OMA lines significantly differ according to ELWL test results, which implies some anatomical and physiological adaptation to water loss from tissues. On the first day of drought, plants possessed 34% more soluble sugars compared to control plants. On the fourteen day of drought, the amount of soluble sugars was reduced by 41.2%. A significant increase of phenolic compounds was observed in the fourteen day of drought, an average of 6%, even up to 57% in line 9. Soil drought substantially reduced stem biomass, grains number, and mass per plant. Lower water loss revealed by results of the ELWL test correlated with the high yield of OMA lines. Phenolic compound content might be used as a biochemical indicator of plant drought tolerance since there was a significant correlation with the high yield of plants subjected to soil drought.

## 1. Introduction

Drought is acknowledged as one of the most important factors lowering agricultural yield, including crops such as wheat, rice, and maize [1]. Depending on the severity of the stress, the type of crop, genetic characteristics, the presence of other stressors, and the stage of development of the plant, the plant’s response to drought stress can vary [2].

One of the plant’s defenses is soluble sugars. Single carbohydrate molecules are simple sugars, such as glucose or fructose. When one carbohydrate molecule is made up of several smaller ones, we are talking about oligosaccharides, such as sucrose. Polysaccharides, such as starch, glycogen, or cellulose, are polysaccharides composed of many sugar molecules. Each type of carbohydrate plays an important role in plants, such as an energy reserve, performing signaling or regulatory functions, influencing the maintenance of turgor, or causing the opening and closing of stomata [3]. In conditions of increasing water scarcity, the efficiency of all processes in the plant is at risk. Plants that are more tolerant to drought stress do better when intracellular pressure drops due to water loss. It depends to a large extent on the concentration of cell sap in the cytosol. Its key components are soluble sugars, the accumulation of which is correlated with resistance to drought stress [4]. Accumulation of sugar reserves as preparation of plants for the stress of water shortage has been observed in many very different species, such as grasses or potatoes [5,6]. Storing large amounts of carbohydrates can inhibit photosynthesis while stimulating respiration. In addition, their stock can be used after the drought subsides as a source of readily available carbon [7]. The concentration of soluble sugars in the cell sap is a direct response to a given stressor. The functions of some sugars may be responsible for indirect reactions, such as inducing the expression of some genes responsible for plant resistance or being precursors of many defense substances synthesized in the plant as a result of stress. The role of soluble sugars in the plant is also extremely important in the case of biotic stresses. By influencing changes in the water potential in cells infected with the pathogen, they limit its development. For example, higher resistance of grasses to snow mold was demonstrated at low water content and high concentrations of soluble sugars in cells [8,9]. As a result of the stress factor, the synthesis of various types of compounds is induced in plants, which participate in the proper reaction aimed at limiting its harmful effects [10].

Another large group of molecules generated by plants in order to fight the stress factor are phenolic compounds. The amount of phenolic compounds in plants changes during the growing season and depends largely on the stress factors. It is a very diverse group of secondary metabolites divided into simple and complex phenols, which are derivatives of phenols. They have a protective function against herbivores and affect the taste and smell of plants. Phenolic compounds in root secretions have a significant effect on the phenomenon of allelopathy. The scopoletin secreted by oat plants interrupts the reproductive cycles of many pests and diseases of cereals. For this reason, oats are considered a phytosanitary plant, which improves its position in cereal rotations [11]. In turn, phenylpropanoid polymers, such as lignin and suberin, saturate cell walls. A large group of complex phenols are tannins and flavonoids. They mainly have protective functions, being poisons and repellents that protect against pathogens and pests. The toxic properties of phenols include the denaturation of protein. Anthocyanins, flavonols, and flavones are the individual groups of flavonoids that make up the known groups of phenolic compounds. Increased content of phenolic compounds as a plant response to stress also contributes to the neutralization of reactive oxygen species (ROS), which are often the fastest response and effect of a stressor [12].

In crop species, there are several instances of sexual hybridization where stable, viable hybrids have been produced, such as the oat (*Avena sativa* L. 2n = 6x = 42) and maize (*Zea mays* L. 2n = 2x = 20), which are particularly intriguing since they are clearly related plant species that are capable of sexual reproduction [13]. The discovery of keeping corn chromosomes in pollinated oat plants led to the development of the oat × maize addition (OMA) lines [14] and their use in streamlining the study of the maize genome [13]. Numerous applications of OMA lines in research might be enumerated as follows: investigations on the expression of maize genes in the oat genetic background, including looking at elements of gene regulation [15], the potential acquisition of novel traits [16], and disease resistance [17], but also abiotic stress like drought [18]. The OMA lines are crosses between plants with C3 and C4 photosynthesis, allowing researchers to better understand C4 photosynthesis [16]. In order to follow the genetics of the maize C4 pathway and identify chromosomes/chromosome regions significant in this process, OMA lines may also be used. The C4 photosynthesis is thought to be more efficient, and plants that perform that type of CO_2_ assimilation are more resistant to photooxidation [16,19,20]. Other uses of OMA lines have been noted in the field of molecular genetics, including investigations into the structure of the maize centromere and knob [10], the behavior of chromosomes during meiosis [17], the use of fluorescence in situ hybridization (FISH) to physically map single-copy sequences on maize chromosomes [21,22], and the procedure of flow-cytometry to separate individual maize chromosomes [23]. Additionally, the presence of maize chromatin frequently indicates morphological (such as thickened shoots, straight leaf blades, and bent panicles) and physiological (such as abnormal panicle growth and chlorophyll production) anomalies, but their characteristics vary depending on the specific addition of maize chromosomes as well as oat genetic background [19]. The greater tolerance of the OMA lines to different stressors, including resistance to environmental variables, such as *Puccinia coronata* f. sp. *avenae* or *Puccinia graminis* f. sp. *avenae*, is therefore assumed, as well as their effect on the behavior of the photosynthetic apparatus [24,25,26].

Oat is a very important cereal crop due to its high nutritional value, unique biological properties, and use in the cosmetics sector [27]. With a seeded area of 9.77 million hectares and a total harvest of 25.18 million tons in 2020, the average yield was 25.77 dt ha^−1^. Oats are mostly farmed in Australia, Canada, and Russia. In 2020, these three nations cultivated 45% of all agricultural land worldwide. Canada topped Russia in terms of harvest volume (4.57 million t vs. 4.13 million t, respectively) thanks to higher oat grain yields [28]. Drought can diminish cereal yields by lowering the number of viable panicles and the number of grains per year and disrupting grain filling (lower thousand-grain weight) [27]. However, oats exhibit increased vulnerability to drought stress and require more water during the vegetative phase than other cereal crops [29]. When being sown in early spring, spring cereals require a longer time than winter cereals to reach the crucial stage of shooting at the stem, which increases the risk of drought-related losses due to a shortage of rainfall when the water stores in the soil runs out [30]. The pouring of the grain is another crucial growth stage because a shortage of water at this stage causes the grain to fill poorly, resulting in a reduction in the weight of a thousand grains. Both winter and spring cereals fall under this [31]. In regions where late spring droughts are common, plant breeding produces early crop types that begin the most important water-demanding stages of the cycle as soon as feasible to get through them before protracted drought conditions arise [32]. Studies on the impact of dryness during grain filling on winter barley reveal a production decline of even more than 80% in both greenhouse and field circumstances [33]. Drought may spread to even more locations as a result of climate change, and its impacts may be more severe [34].

The first aim of the research was to identify OMA lines among plants generated by wide crossing with maize as a pollinator, then to investigate possible differences in OMA lines according to the Excised Leaf Water Loss (ELWL) test, selected biochemical parameters and agronomic characteristics under water availability and simulated soil drought conditions. Also, detection of a similar response to soil drought stress occurred in subsequent OMA generations (F2 and F3). The applied goal was to choose the lines best suited to development and cultivation under water limitation based on the results of the aforementioned features.

## 2. Results

### 2.1. Molecular Identification of OMA Lines

Based on the results of electrophoretic separations, introgression of maize genetic material was found in 15 descendants resulting from wide crossing of oat with maize. Out of 120 tested genotypes, the rest were oat doubled haploids (DH) lines. Therefore, it was detected that 12.5% of plants generated as a result of oat pollination with maize were OMA lines (oat maize addition). The OMA lines with the Grande 1 retrotransposon fragment (500 bp) presence are annotated as follows: 1b, 9, 12, 18, 23, 26, 35, 42, 43, 55, 78b, 83, 97, 114 and 119. The example result of DNA separation is presented in Figure 1.

### 2.2. Excised Leaf Water Loss Test and Biochemical Analysis

All the traits had a normal distribution. The results of multivariate analysis of variance (MANOVA) indicated that treatment, generation, genotype, and all interactions were statistically significant (*p* < 0.001) when examined in all ten quantitative traits jointly. Analysis of variance revealed that the main effects of all factors and all interactions were significant for all the traits of the study, except for differences between control and stress for phenolic compounds content (*p* = 0.818).

The mean water loss of all genotypes after 6 h averaged 28.6%, and ELWL_0–6h_ values ranged from 20.8% to 42.1%. Genotypes 18, 119, and 35 lost the most water and 23, 9, and 12 the least. Detailed results of each tested genotype are presented in Table 1.

On the first day of drought, the plants subjected to water restriction possessed, on average, 34% higher content of soluble sugars compared to the plants normally watered. A statistically insignificant decrease occurred only in line no. 18 and in the Bingo variety. The remaining lines showed an increase, and in nine genotypes, it was statistically significant. The highest increases of 70% were observed in lines no. 43, 12, 9, and 1b. The highest contents of soluble sugars in the control plants were recorded in lines no. 23 and 35 and in the Bingo variety, while the lowest contents were found in lines no. 43 and 42. Among the OMA lines in drought stress, the highest contents of soluble sugars were found in lines no. 18 and 42 (Table 2).

On the fourteenth day of drought, a decrease in the content of soluble sugars was noted compared to the first day of drought. In three genotypes and in the variety Bingo, the level of soluble sugars in combination with drought was lower than in the control plants, but significant decreases occurred only in line no. 78b and in the variety Bingo, where they amounted to 34% and 39%, respectively. In 11 OMA lines, an increase in the content of soluble sugars was observed as the effect of drought, but significant only in six lines, and the highest, over 100%, was observed in lines no. 18 and 1b. Among the control plants, the highest content of soluble sugars was recorded in lines no. 23 and 119 and in the Bingo variety, while the lowest in lines no. 42 and 18. In the group of plants subjected to drought stress, the highest contents were found in lines no. 83 and 1b and the lowest in no. 42 and 78b (Table 2).

On the first day of drought, there were no statistically significant differences in the average content of phenolic compounds between control plants and plants in drought conditions. However, significant differences were observed between individual genotypes, such as in line no. 119, in which an increase in the content of phenolic compounds due to drought stress was observed, and in line no. 1b a decrease compared to control plants. Statistically significant differences as a result of drought occurred only in these two genotypes. Among the control plants, the highest content of phenolic compounds was observed in lines no. 43 and 42, however the lowest in lines no. 23 and 119 and in the Bingo variety. In the plants subjected to drought, the highest content of phenolic compounds was recorded in lines no. 114 and 43, however the lowest in lines no. 1b and 55 and in the Bingo variety (Table 3). On the fourteenth day of drought, significant differences in the content of phenolic compounds between control plants and plants growing in drought conditions were observed. On average, 6% more phenolic compounds were found in plants subjected to drought compared to control plants. As a result of drought, the highest increase in the content of phenolic compounds (57%) was recorded in line no. 9, and the only significant decrease occurred in line no. 114 and amounted to 21%. In the control conditions, the highest concentrations of phenolic compounds were recorded in lines no. 42 and 114, however, the lowest in lines no. 18 and 26. In drought conditions, the highest concentrations of phenolic compounds were recorded in lines no. lines no. 18 and 35 (Table 3).

### 2.3. Analysis of Primary and Secondary Shoots Biomass and Selected Yield Elements

Significant differences in the total mass of shoots were found between control and drought plants. On average, as a result of drought, 43% less biomass of aboveground parts was collected. No increase in shoot weight due to drought was found in any of the genotypes. Only for line no. 114, the decrease in the total weight collected was not statistically significant. In the remaining OMA lines and in the Bingo cultivar, the decreases in the harvested biomass of aboveground parts due to drought were statistically significant. The highest decreases were found in lines no. 18 and 78b and in the Bingo variety, where they amounted to 58%, 56%, and 75%, respectively. Among the control plants, the highest shoot weights were recorded on lines no. 78b and 1b, however, the lowest on lines no. 114 and 83. On the other hand, when considering a combination with drought, the highest masses of shoots were found in lines no. 1b and 119, however, the lowest in lines no. 83 and 18 and in the Bingo cultivar (Table 4).

The number of developed grains differed in particular genotypes. As a result of drought, there was a statistically significant decrease in the number of grains in drought combination by an average of 31% compared to control plants. Statistically significant decreases in the number of grains occurred in lines no. 23, 78b, 83, and 9 and in the Bingo variety, where they amounted to 52%, 44%, 38%, 25%, and 62%, respectively. Among the control plants, the most grains were found in lines no. 78b and 9, however, the least in lines no. 114 and 18. Whereas in drought conditions, the most grains were found in lines no. 9 and 78b, the least in lines no. 18 and 114 (Table 4).

The mass of grains developed by a single plant decreased in combination with drought by an average of 34% compared to control plants. Only in three lines, statistically insignificant increases in grain mass were found. This was the case with an overall very low number of developed grains. In the remaining genotypes, decreases were noted. In lines no. 1b, 83, 78b, and 9, and in the Bingo variety, they were statistically significant. Among the plants of the control combination, the highest weights of grains were recorded in lines no. 9 and 78b and in the Bingo variety; however, the lowest was in lines no. 18 and 114. Among the plants subjected to drought, the highest weights were found in lines No. 9 and 78b and the lowest were found in lines 18 and 114 (Table 4).

### 2.4. Correlations among Biochemical Parameters, Biomass and Yield Elements

Soluble sugar content on the first day of drought (20% of soil field capacity) was significantly negatively correlated with phenolic compound content on the first day of drought in both control (−0.70) and drought stress (−0.39) (Figure 2 and Figure 3, respectively). Similarly, for both treatments, control and stress, significant positive correlations were observed between the number of grains and the mass of grains plant^−1^ (0.98 and 0.99, respectively) (Figure 2 and Figure 3). In addition, under control conditions, correlations were observed between ELWL after 0–3 h and ELWL after 4–6 h, ELWL after 0–3 h and ELWL after 0–6 h, ELWL after 4–6 h and ELWL after 0–6 h, soluble sugars content in the first day of drought and soluble sugars content after two weeks of drought, phenolic compounds content in the first day of drought and phenolic compounds content after two weeks of drought, Moreover, a positive correlation between the mass of stems plant^−1^ and the number of grains as well as negative between phenolic compounds content in the first day of drought and soluble sugars content after two weeks of drought were observed (Figure 2).

In addition, under stress conditions, a positive correlation was observed between five pairs of traits: soluble sugars content in first day of drought and the mass of stems plant^−1^, soluble sugars content in first day of drought and the number of grains, soluble sugars content in first day of drought and the mass of grains plant^−1^, phenolic compounds content after two weeks of drought and the number of grains and phenolic compounds content after two weeks of drought and the mass of grains plant^−1^ (Figure 3).

Each trait was of varying significance and had different shares in the joint multivariate variation in the examined genotypes. Analysis of the first two principal components for studied genotypes is shown in Figure 4 (for control) and Figure 5 (for drought stress). In the graphs, the coordinates of the point for studied lines and cultivar Bingo are the values for the first and second principal components. For the control condition, the first two principal components accounted for 90.50% of the total variability between the individual genotypes (Figure 4). The most significant (loading factors) positive linear relationship with the first principal component was found for soluble sugar content on the first day of drought (0.980) and soluble sugar content after two weeks of drought (0.737), whereas the most significant negative linear relationship was found for phenolic compounds content in the first day of drought (−0.701). The second principal component was significantly (loading factors) negatively correlated with the number of grains (−0.958) and the mass of grains plant^−1^ (−0.942).

For drought stress, the first two principal components accounted for 94.04% of the total variability between the individual OMA lines and cultivar Bingo (Figure 5). The most significant (loading factors) positive linear relationship with the first principal component was found for soluble sugar content on the first day of drought (0.945) and soluble sugar content after two weeks of drought (0.507), while the most significant negative linear relationship was found for phenolic compounds content in the first day of drought (−0.430). The second principal component was significantly (loading factors) positively correlated with the number of grains (0.574) and the mass of grains plant^−1^ (0.620), whereas it was significantly negatively correlated with soluble sugar content after two weeks of drought (−0.858).

Excised-leaf water loss after 0–3 h (observed in control only) determined the mass of stems plant^−1^, the number of grains, and the mass of grains plant^−1^ observed in drought stress (Table 5).

In all three cases, the impact was inversely proportional, and the coefficient of determination was not large, ranging from 2.6 (for the mass of grains plant^−1^) to 3.2 (for the number of grains). The mass of stem plant^−1^ was determined by ELWL after 0–6 h, with values explained at 3.0% (Table 5). OMA lines are mostly oat-shaped, but many of the genes contained on the retained maize chromosomes are expressed, which may contribute to the phenotype of the OMA lines. In Figure 6, we would like to report the visual appearance of the OMA plants and control genotypes possessing a pure oat genome. i.e., cv Bingo, under optimal water conditions and subjected to soil drought stress.

## 3. Discussion

When certain related species are crossed, the pollinator’s chromosomes might be totally excluded, resulting in haploid offspring generation. This method, widely known as interspecies or intergeneric crossing (wide crossing), is utilized in plant breeding to speed up breeding activities by obtaining a totally homozygous generation [35]. When the above-mentioned technique is applied in the crossing of oat with maize, certain maize chromosomes are retained in embryogenesis during mitotic cell divisions, and they act like oat chromosomes and are permanently integrated into the genome of newly produced hybrids. OMA lines (oat × maize addition lines) can be detected with molecular methods application, e.g., PCR with specific retrotransposon Grande I primer application. The method was used by many authors to detect OMA lines [26,36]; the percentage of OMA lines could be high, even 47% [36], but in our study in the group of 120 offspring plants resulted from wide crossing oat with maize, 12.5% OMA lines were detected. OMA lines could be useful genotypes in plant breeding, but they also aid in the mapping of the maize genome [37].

The water loss test (ELWL) performed in our experiment revealed significant differences between the tested genotypes. Different crops and, within them, different genotypes having low values of ELWL have a greater ability to maintain water balance in the leaves, largely due to soluble sugars in the plant [38,39]. The combination of low water loss and high content of soluble sugars allows for maintaining water balance during stresses and provides such genotypes with tolerance to drought stress, in effect, higher yield stabilization [39]. In our experiment, line no. 23 was characterized by the lowest water loss among all genotypes and was also characterized by the highest content of soluble sugars, especially compared to other drought-stressed genotypes. Negative correlations between the percentage of water loss and the analyzed yield components indicate genotypes that cope better with soil drought stress. If water loss is lower, yield under drought stress conditions is higher. On this basis, potentially valuable genotypes can be selected for recombinant breeding aimed at creating varieties tolerant to drought stress.

In our experiment, the content of soluble sugars in oat plants increased as a result of soil drought. However, there were significant differences among tested genotypes, which was proved by a significant interaction of two experimental factors, T × G (Treatment and Genotype), both on the first day and fourteenth day of drought. During prolonged drought conditions, the content of soluble sugars decreased (comparison of the first and fourteenth day of drought), but it was still higher in drought-treated plants than in control plants. The accumulation of soluble sugars in plants has been found in many species as a reaction to drought stress, e.g., in maize, oats, rapeseed, or rice, in which the reaction to salinity stress was additionally studied [40,41,42,43]. The beneficial effect of soluble sugars during persistent water shortages was basically manifested by regulating the osmotic pressure of plants, which prevents significant decreases in turgor. Sinay and Karuwal [44], in a greenhouse experiment with nine corn varieties, investigated the effect of watering intervals on the content of soluble sugars [45]. In the control, all plants were watered every other day, while the conditions of water deficiency were induced on two levels: watering only every 8 or 12 days. As a result of the induced drought conditions, significant increases in the content of soluble sugars in maize leaves were found, depending on the genotype, several times to several hundred times compared to the values in control plants [44]. In addition, a significant positive correlation was found between the content of soluble sugars and the analyzed yield components.

The content of phenolic compounds is another possible parameter applied in our studies to indicate genotypes tolerant to soil drought stress. During the experiments, it increased in both control and drought plants when compared to the first and the fourteenth day of drought. In most cases, the values due to stress were higher. On both dates (the first and fourteenth day of drought), the genotype had a significant impact on the content of phenolic compounds, which indicates a different reaction of the tested OMA lines to water deficiency. In experiments on drought stress in spring triticale, an increase in the content of phenolic compounds was found, and this parameter was considered useful in selecting genotypes less susceptible to drought stress [46]. In potatoes, increased content of phenolic compounds was also observed as a result of drought, during which the analyzed secondary metabolites were responsible for the good general condition of plants [47]. The positive effect of increased phenolic compounds content as antioxidants supporting plants in the fight against ROS (reactive oxygen species) induced by stress has also been demonstrated in other species, such as rapeseed or potato [44,47].

The last analyzed parameters were aboveground biomass and selected yield elements. The occurrence of a two-week period of drought during the growing season of oat-maize hybrids affected the total dry mass of shoots, as well as the number and mass of grains. Genotype, treatment, and their interaction had a significant impact on the differences. The average decrease in the total aboveground mass was 43%. In the most susceptible genotypes, the decrease reached 75%, and in the tolerant ones, the differences were not statistically significant. The experiments also showed a significant reduction in the number and weight of grains. Among all analyzed lines, the highest values of yield components were achieved by lines no. 9 and 78b, especially in the case of the number and weight of grains. The total biomass of aboveground parts of these lines decreased as a result of drought, but it was one of the smallest percentage decreases in these values compared to the rest of the tested genotypes. These lines were the only ones among the 14 analyzed OMA lines that achieved higher or similar values to the Bingo variety in the control and even exceeded the standard object (cv. Bingo) in drought conditions. None of the genotypes reached a higher number and weight of grains under drought stress when compared to lines no. 9 and 78b. Bingo variety reached similar values only for some yield elements. Studies conducted on wheat showed lower biomass values in plants growing in drought conditions compared to control ones. The decrease in yield was 59% for fresh weight and 51% for dry weight [48]. Other studies on drought stress in barley revealed a significant decrease in the number of grains as well as their weight due to drought in both mild and severe drought stress. As a result of drought, the grain filling period was shortened by as much as ⅓ in the case of severe drought stress. The pouring of grain ended earlier, which caused a shortening of the vegetation period and led to a situation where, 17 days after the beginning of this phase, the average weight of one grain was higher in drought-stressed plants than in control plants. Grain pouring in the control plants continued after 17 days from the beginning of this phase, so much higher grain weights than those in drought conditions were reached only after the end of drought vegetation [49]. In our experiment, earlier maturation of plants subjected to drought stress was also observed. Higher production of biomass and a higher number and weight of grains were associated with higher contents of soluble sugars on the first day of drought or higher contents of phenolic compounds on the fourteenth day of drought. These findings were confirmed by significant correlations.

These associations might find possible application in crop improvement programs since the excised leaves water loss (ELWL test), as well as the amount of phenolics and sugars, significantly correlated with the yield of plants under soil drought. These physiological and biochemical features can be recommended in plant breeding as a fast screening test of the germplasm possessing higher soil drought tolerance.

## 4. Materials and Methods

### 4.1. Molecular Identification of OMA Lines

The plant material for the study was obtained by wide crossing of oat with maize at The Franciszek Górski Institute of Plant Physiology, Polish Academy of Sciences in Krakow. Intraspecific oat F1 hybrids (cross combinations of cultivars or breeding lines) pollinated with maize allowed to generate a population of DH lines and OMA lines equivalent to the classic F2 generation, but with the difference that only homozygous forms constituted this population [36,50]. A total of 120 descendants of wide crosses were tested to detect oat-maize hybrids. The control plants were maize variety Waza (positive control) and oat variety Stoper (negative control). Identification of hybrids was carried out using the PCR method, with the application of the marker retrotransposon fragment sequence Grande 1 present in multiple copies on each maize chromosome [23]. The leaves of plants grown in controlled greenhouse conditions were used for the study.

### 4.2. Grande I Amplification

The harvested oat leaves were lyophilized under reduced pressure (40 μbar) at a heat exchanger temperature of −52 °C (FreeZone 6 L lyophilizer, Labconco, Kansas, MO, USA). The plant material was then homogenized using a cryogenic homogenization mill (MM400, Retsch, Haan, Germany) for 5 min at a frequency of 25 Hz in 2000 μL round-bottom tubes together with 5 grinding balls made of zirconium oxide (Ø5 mm). Genomic DNA isolation was performed using the Genomic Mini AX Plant enhanced performance kit from A&A Biotechnology; the DNA was then dried in a concentrator (Concentrator Plus, Eppendorf, Germany) for 10 min at 20 mbar. Then, dried DNA samples were diluted in TE buffer (Tris, EDTA) for 48 h. The concentration of the DNA solution was measured using the NanoDrop 2000 c spectrophotometer (Thermo Scientific, Waltham, MA, USA). Finally, the DNA concentration of 50 ng μL^−1^ was used in the PCR reaction. Apart from the OMA lines samples, the maize DNA was isolated from cv. Waza (positive control) and oat DNA from variety cv. Stoper (negative control) was used in the reaction. The reaction mixture contained 15.5 μL of water, 3 μL of MgCl_2_ solution, 2 μL of free nucleotides (dNTP) solution, 2.5 μL of Taq 10× buffer, 1 μL of Grande 1R primer solution, 1 μL of Grande 1F primer solution, 0.42 μL of Taq polymerase (Thermo Scientific, USA). The primers used bound to highly conserved elements, such as the Grande 1 retrotransposons, which are common on each maize chromosome [14] (GenBank accession number X976040. The Grande 1 primer sequence was as follows:

GRANDE 1F: 5′-AAAGACCTCACGAAAGGCCCAAGG-3′

GRANDE 1R: 5′-AAATGGTTCATGCCGATTGCACG-3′

The PCR reaction was carried out under the following conditions: pre-denaturation −94 °C for 5 min; 25 cycles: denaturation −94 °C for 30 s, primer annealing −58 °C for 30 s, polymerization −72 °C for 30 s; final polymerization at 72 °C for 5 min and cooling the samples to 4 °C. Separation of the amplification products according to the length of the newly formed DNA fragments was carried out by electrophoresis in 1.5% agarose gel in TBE buffer (Tris, boric acid, EDTA—Sigma Aldrich, St. Louis, MO, USA) at a voltage of 110 V for 60 min. Archiving of electrophoretic separations was made using a gel archive system (Amersham—Pharmacia Biotech, Piscataway, NJ, USA) and the Liscap Capture Application ver. 1.0. The analysis of the obtained PCR products was performed using the GelScan ver. 1.45 (Kucharczyk—Electrophoretic Techniques, Warsaw, Poland). Computer analysis of the gels allowed the detection of oat DH lines and OMA lines.

### 4.3. Greenhouse Experiment

OMA lines that were chosen based on molecular analyses were replicated to create F3 offspring, which were then evaluated in a greenhouse experiment in the spring of 2020. The Bingo oat variety was used as a reference genotype, and two generations (F2 and F3) of fourteen OMA lines (No. 1b, 9, 12, 18, 23, 26, 35, 42, 43, and 55) were evaluated. The plants were given a soil drought treatment, while the control group consisted of the same plants that were consistently watered during the trial. In Krakow’s Department of Plant Breeding, Physiology, and Seed Science, the experiment was conducted in the greenhouse. Four replications of the experimental factor—i.e., drought conditions and optimal soil water content—were prepared for each OMA line. The weather conditions for the subsequent months of vegetation (April to August) were as follows: temperature: 4.7, 9.3, 11.3, 18.0, 18.9, and 20.3 °C, relative humidity: 64.0, 48.5, 68.5, 77.7, 69.7 and 69.2%.

A total of 240 pots were used in the experiment (15 genotypes, including 14 OMA lines and the cv. Bingo; four replicates; two generations; and two treatments). In February 2020, the genotypes under study had their seeds planted. The pots held a total of 2500 g of soil and sand combined in equal parts. In order to simulate a drought, watering was restricted after the soil humidity level reached 20%. The control pots received regular watering to 70% of soil humidity throughout the simulation of drought conditions, whereas the pots under the stress of the drought received only water to a moisture level of 20%. Soil water content was controlled using a HydroSense^®^ Soil Water Measurement System 620 (CAMPBELL SCIENTIFIC Inc., Shepshed, Leicestershire, UK) affixed with two 12-cm long sensors.

The following experiments and activities were conducted during the greenhouse experiment:Performing ELWL (Excised Leave Water Loss)Collection of leaves for soluble sugars and phenolic compound content on the first day of reaching 20% humidity simultaneously with a control combination (70% soil humidity).Collection of leaves for soluble sugars and phenolic compound content on the fourteenth day of drought (maintaining 20% humidity) simultaneously with a control combination (70% soil humidity).Harvesting mature shoots, weighing the biomass of aboveground parts and total mass of grains from all shoots.

### 4.4. Water Loss Test

A physiological indicator widely used as a selection criterion for drought tolerance is the rate of Excised Leaf Water Loss (ELWL), which in soil drought conditions is negatively correlated with grain yield [51]. The water loss test was performed on OMA lines grown under controlled greenhouse conditions. The cut leaves of oat plants at the BBCH 27 development stage (end of tillering) were transferred to the growth chamber, where the following measurement conditions were maintained: temperature 20 °C, air humidity 50%, and lighting 250 µmol m^2^ s^−1^ (HPS “Agro” lamps, Osram). Water loss was monitored by weighing the leaves immediately after cutting (0 h), after 3 and 6 h, and after drying at 70 °C for 48 h. The water loss in the tested plants was calculated on the basis of the formula given by Clarke and McCaig [52]:ELWL_0–3h_ = (FW0 − FW3)/(FW0 − DW),ELWL_4–6h_ = (FW3 − FW6)/(FW3 − DW),ELWL_0–6h_ = (FW0 − FW6)/(FW0 − DW),

where: FW0—fresh mass, FW3—fresh mass after 3 h, FW6—fresh mass after 6 h, DW—dry weight after drying for 48 h at 70 °C.

### 4.5. Analysis of Biochemical Parameters

The leaves for biochemical analysis were collected twice during the experiment: on the first day of drought (when the soil reached 20% of capacity) and on the fourteenth day of drought (the maintenance of soil 20% capacity), both from control plants and plants subjected to soil drought. One leaf was collected from the main shoots of each of the 3 pots of the F2 generation and 3 pots of the F3 generation. The harvested oat leaves were lyophilized under reduced pressure (40 μbar) at a heat exchanger temperature of –52 °C (FreeZone 6 L lyophilizer, Labconco, USA). The plant material was then homogenized using a cryogenic homogenization mill (MM400, Retsch, Germany) for 5 min at a frequency of 25 Hz in 2000 μL round-bottom tubes together with 5 grinding balls made of zirconium oxide (Ø5 mm). For biochemical measurements, 5 mg of lyophilized and homogenized leaves were utilized with the addition of 2 mL of 80% ethanol; then, the tubes were centrifuged at 2800 rpm for 20 min (Eppendorf Centrifuge 5702 R, Eppendorf, Hamburg, Germany). The total amount of sugars was determined by the phenol method by Dubois et al. [53]. The reaction mixture contained 0.2 mL of distilled water, 20 μL of supernatant, 0.2 mL of 5% phenol, and 1 mL of concentrated H_2_SO_4_. After 10 min of mixing the reaction components, the absorbance was measured at a wavelength of 490 nm (Synergy 2 spectrophotometer, BioTek, Winooski, Vermont, USA). The content of sugars was calculated in mg of sucrose per 1 g of dry matter of plant tissue [mg g^−1^ DM]. The content of phenolic compounds was determined by the method of Folin and Ciocalteu [54]. The reaction mixture contained 1 mL of water, 20 μL of supernatant, 0.5 mL of 25% Na_2_CO_3_, and 0.125 mL of Folin and Ciocalteu reagent (diluted 1:1 with distilled water shortly before use). After 30 min, the absorbance was measured on a spectrophotometer (Synergy 2, BioTek, USA) at a wavelength of 760 nm. The content of phenolic compounds was determined in mg of chlorogenic acid per 1 g of dry matter of plant tissue [mg g^−1^ DM].

### 4.6. Analysis of Primary and Secondary Shoots Biomass and Selected Yield Elements

At the stage of grains’ full maturity, all plants of control and subjected to drought stress conditions were harvested. The primary and secondary shoots biomass generated by plants was assessed, as well as the percentage of grains in the overall biomass yield. It was noted the quantity of shoots, their weight, and the weight of the grains. Each shoot was measured independently, and the total amount of branching on each individual plant was then calculated, and the results were gathered for each individual plant as described by Warzecha et al. [18].

### 4.7. Statistical Analysis

The normality of distribution of the ten traits was tested using Shapiro–Wilk’s normality test [55,56] to verify whether the analysis of variance (ANOVA) met the assumption that the ANOVA model residuals followed a normal distribution. Three-way multivariate analysis of variance (MANOVA) was performed. Three-way analyses of variance (ANOVA) were carried out to determine the main effects of treatment, generation, and genotype, as well as their interactions on the variability of the particular traits. The mean values and standard deviations of traits were calculated for treatments, generation, genotypes, and their combinations. Additionally, Fisher’s least significant differences (LSDs) were calculated for individual traits at the 0.05 level, and on this basis, homogeneous groups were generated. The relationships between observed traits were estimated using Pearson’s linear correlation coefficients based on the means of the genotypes, independent of control and drought stress. The relationships of the observed traits are presented in heatmaps. The results were also analyzed using multivariate methods. Principal component analysis (PCA), independent of control and drought stress, was utilized to show a multi-trait assessment of the similarity of the tested genotypes in a lower number of dimensions with the least possible loss of information [57]. The effect of excised-leaf water loss values, independent: after 0–3 h, after 4–6 h, and after 0–6 h, on the mass of stems plant^−1^, the number of grains, and the mass of grains plant^−1^ was assessed by regression analysis. All these analyses were conducted using the GenStat v. 23 statistical software package [58].

## 5. Conclusions

Among the 120 tested plants generated by wide crossing oat with maize, the presence of 105 DH (doubled haploids) lines and 15 OMA (oat × maize addition) lines was confirmed, which is 12.5% of all tested lines. The analysis of variance showed a significant effect of soil drought on the content of soluble sugars on the first day of soil drought, and on the fourteenth day of soil drought, a significant effect of stress on the content of soluble sugars and phenolic compounds was noted. Moreover, drought caused a significant decrease in the aboveground biomass of plants and selected yield elements. The tested OMA lines had a significant impact on the values of the excised leaf water loss (ELWL) test, as well as biochemical parameters and selected yield components in response to drought stress. In the entire experiment, lines no. 9 and 78b were the most distinctive, as they were the only ones among the analyzed genotypes to achieve higher values of the tested yield components than the Bingo cultivar, both in the control and in the soil drought conditions. It is worth noting that for the OMA line no. 9 significant correlation between high yield values and low water loss from the leaves (ELWL test) was found. The ELWL values classified line no. 9 among the three genotypes with the lowest water loss. Therefore, it can be concluded that this genotype is more tolerant to drought stress. OMA lines no. 9 and 78b should be considered the best candidates for further investigation and possible application in breeding programs on oat resistance to soil drought stress. Moreover, in both lines, a high content of phenolic compounds was found, correlated with a high yield of grains in drought conditions. This parameter can be utilized as a biochemical indicator of oat tolerance to soil drought in oat.

## Figures and Tables

**Figure 1 ijms-24-13905-f001:**
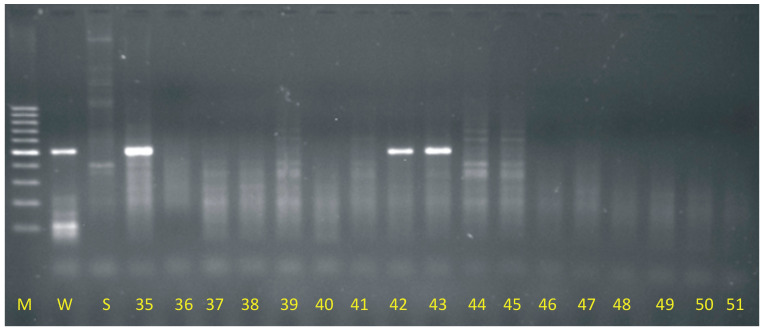
Electrophoretic separation of PCR products generated with Grande 1 primers application (M—100 bp marker, W—maize cv. Waza, S—oat cv. Stoper, 35—OMA line, 36–41—oat DH lines, 42–43—OMA lines, 44–51—oat DH lines).

**Figure 2 ijms-24-13905-f002:**
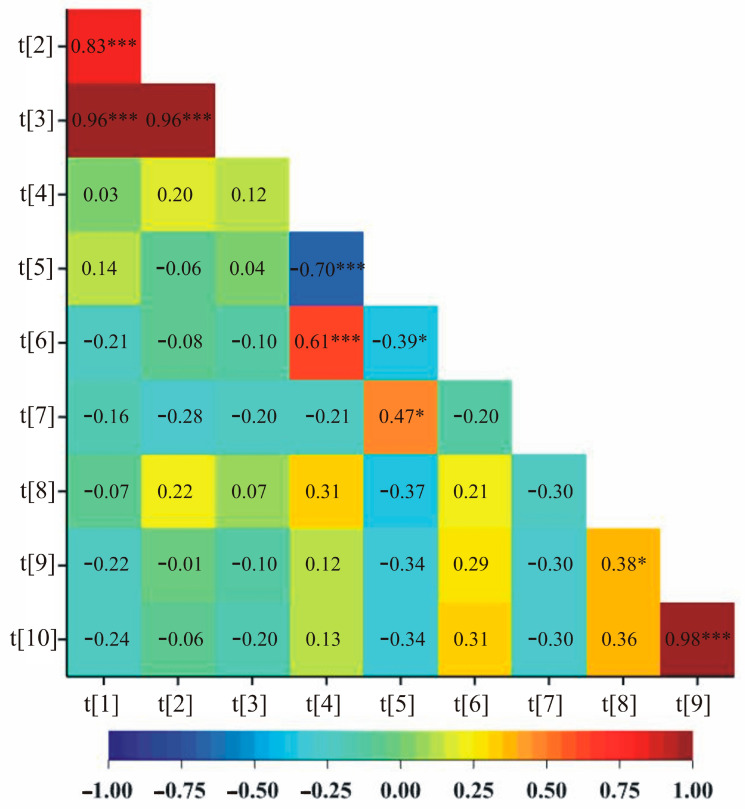
A heatmap showing correlation coefficients between all pairs of observed traits in control. [t[1]—ELWL after 0–3 h, t[2]—ELWL after 4–6 h, t[3]—ELWL after 0–6 h, t[4]—soluble sugar content in first day of drought (20% of soil field capacity), t[5]—phenolic compounds content in first day of drought (20% of soil field capacity), after two weeks of drought (maintaining 20% of soil field capacity), t[6]—soluble sugars content after two weeks of drought (maintaining 20% of soil field capacity), t[7]—phenolic compounds content after two weeks of drought (maintaining 20% of soil field capacity), t[8]—the mass of stems plant^−1^, t[9]—the number of grains, t[10]—the mass of grains plant^−1^]. * *p* < 0.05; *** *p* < 0.001.

**Figure 3 ijms-24-13905-f003:**
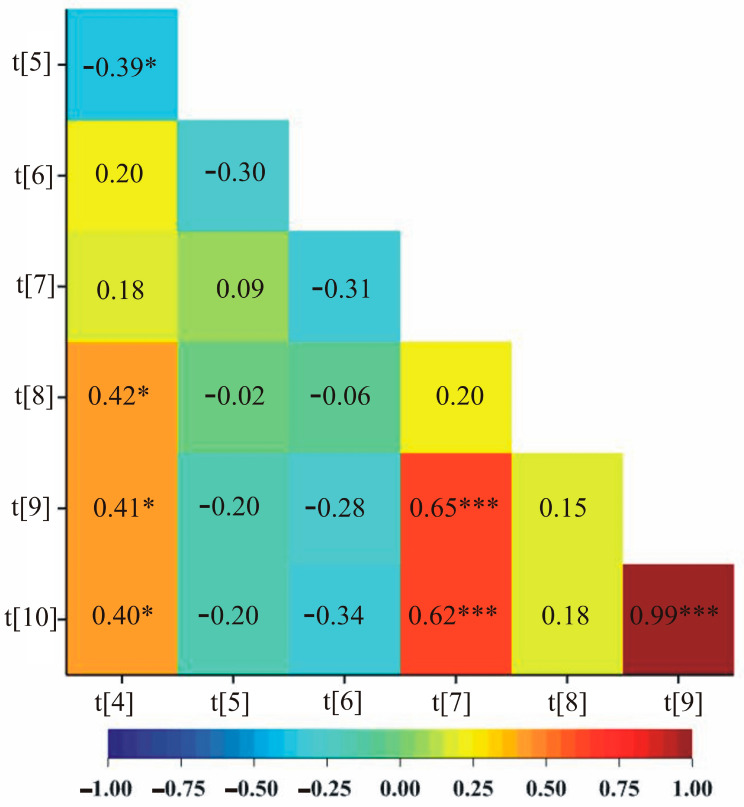
A heatmap showing correlation coefficients between all pairs of observed traits in drought stress. [t[4]—soluble sugars content on first day of drought (20% of soil field capacity), t[5]—phenolic compounds content on first day of drought (20% of soil field capacity), after two weeks of drought (maintaining 20% of soil field capacity), t[6]—soluble sugars content after two weeks of drought (maintaining 20% of soil field capacity), t[7]—phenolic compounds content after two weeks of drought (maintaining 20% of soil field capacity), t[8]—the mass of stems plant−1, t[9]—the number of grains, t[10]—the mass of grains plant^−1^]. * *p* < 0.05; *** *p* < 0.001.

**Figure 4 ijms-24-13905-f004:**
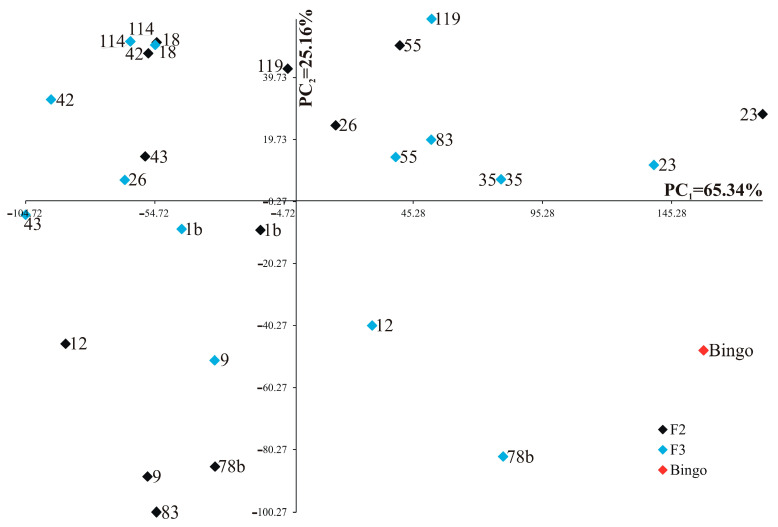
Distribution of fourteen OMA lines in F2 and F3 generations and cultivar Bingo in the space of the first two principal components (PC_1_ = 65.34%, PC_2_ = 25.16%) for control.

**Figure 5 ijms-24-13905-f005:**
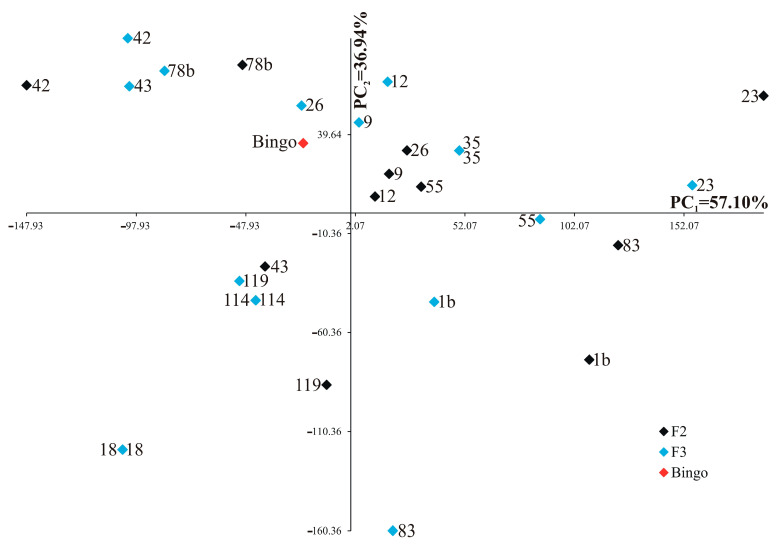
Distribution of fourteen OMA lines in F2 and F3 generations and cultivar Bingo in the space of the first two principal components (PC_1_ = 57.10%, PC_2_ = 36.94%) for drought stress.

**Figure 6 ijms-24-13905-f006:**
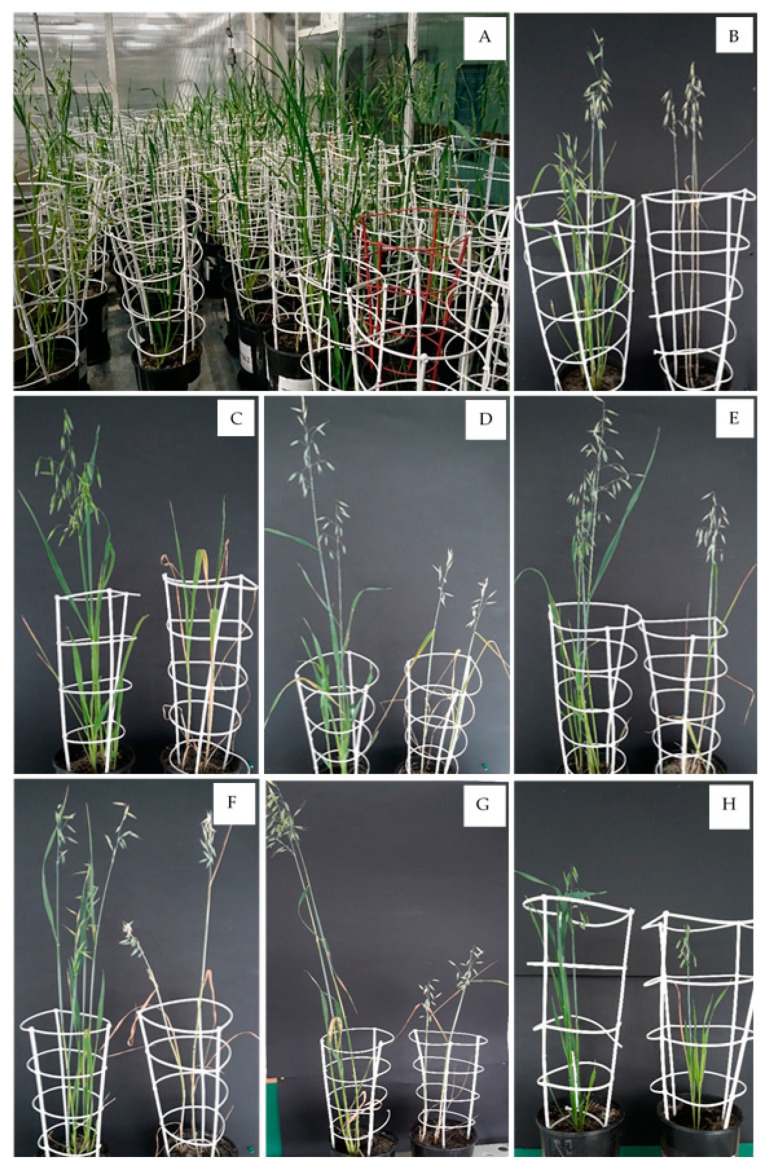
Chosen oat × maize addition lines (OMA) and cv. Bingo: (**A**)—grown in the greenhouse before drought treatment; control plants (**left**) and plants after drought treatment (**right**): (**B**)—cv. Bingo; (**C**)—OMA line 1b; (**D**)—OMA line 9; (**E**)—OMA line 26; (**F**)—OMA line 35; (**G**)—OMA line 43; (**H**)—OMA line 83.

**Table 1 ijms-24-13905-t001:** Mean values, standard deviations, and homogeneous group for excised-leaf water loss.

Trait		Excised-Leaf Water Loss (ELWL)					
		After 0–3 h		After 4–6 h		After 0–6 h	
Genotype		Mean	s.d.	Mean	s.d.	Mean	s.d.
Bingo		19.7 c–g	3.68	14.0 c–e	3.32	30.8 b–e	5.67
F2	114	22.8 bc	5.23	13. 0 c–f	4.27	32.6 b–d	7.85
	119	17.4 e–h	1.49	12.9 c–f	3.04	28.0 b–f	3.55
	12	16.6 gh	2.03	13.7 c–e	11.02	28.0 b–f	9.05
	18	28.3 a	5.77	19.3 b	2.40	42.1 a	6.26
	1b	22.3 b–d	7.37	14.1 c–e	6.47	32.9 b–d	11.08
	23	15.3 gh	3.66	10.3 e–h	4.05	23.8 e–g	6.56
	26	15.4 gh	1.83	10.0 e–h	1.21	23.9 e–g	2.62
	35	21.3 b–f	3.19	16.4 b–d	2.72	34. 1 bc	4.47
	42	16.3 gh	2.91	10.8 e–h	3.34	25.3 e–g	5.30
	43	18.9 c–h	6.30	11.1 e–h	6.23	27.6 c–g	10.28
	55	21.7 b–e	3.93	16.9 bc	4.98	34.8 b	7.02
	78b	15.7 gh	0.82	10.1 e–h	0.75	24.2 e–g	0.11
	83	18.2 d–h	3.52	11.4 e–h	4.06	27.5 c–g	5.89
	9	15.6 gh	2.29	7.6 gh	1.30	22.0 fg	3.20
F3	114	22.8 bc	5.23	13.0 c–f	4.27	32.6 b–d	7.85
	119	16.9 f–h	3.88	11.7 d–h	1.96	26.7 d–g	3.32
	12	17.1 f–h	3.99	9.5 e–h	2.46	24.8 e–g	5.66
	18	28.3 a	5.77	19.3 b	2.40	42.1 a	6.26
	1b	17.2 e–h	2.19	8.5 f–h	3.02	24.3 e–g	4.33
	23	15.3 gh	1.64	8.4 f–h	0.91	22.4 fg	2.12
	26	16.9 f–h	1.48	12.3 c–g	2.54	27.1 d–g	3.35
	35	21.3 b–f	3.19	16.4 b–d	2.72	34.2 bc	4.47
	42	16.3 gh	4.26	10.1 e–h	3.37	24.7 e–g	6.52
	43	16.0 gh	2.69	11.3 e–h	3.18	25.4 eg	5.05
	55	16.7 gh	1.09	12.8 c–f	1.81	27.4 c–g	1.87
	78b	25.6 ab	4.52	26.8 a	6.27	45.3 a	7.68
	83	18.2 d–h	0.50	11.6 e–h	0.75	27.7 c–g	0.17
	9	14.8 h	1.00	7.1 h	1.68	20.8 g	2.28
LSD_0.05_		4.5		4. 8		7.1	

a, b, c, …—In columns, means followed by the same letters are not significantly different.

**Table 2 ijms-24-13905-t002:** Mean values, standard deviations, and homogeneous group for soluble sugars content (mg g^−1^ of d.w.).

Trait		First Day of Drought (20% of Soil Field Capacity)						After Two Weeks of Drought (Maintaining 20% of Soil Field Capacity)					
Treatment		C		S		Average		C		S		Average	
Genotype		Mean	s.d.	Mean	s.d.	Mean	s.d.	Mean	s.d.	Mean	s.d.	Mean	s.d.
Bingo		374.9	19.98	340.1	42.86	357.5 bc	36.96	263.9	77.13	162.1	8.13	213.0 b–g	74.64
F2	114	204.6	28.81	295.8	26.54	250.2 g–l	55.09	139.6	12.26	218.6	10.10	179.1 f–k	43.47
	119	244.0	48.65	305.7	10.88	274.8 e–j	46.38	202.2	32.22	272.6	14.41	237.4 a–c	44.14
	12	160.1	19.47	352.3	114.43	256.2 g–l	127.8	142.5	22.68	206.6	3.88	174.5 g–j	37.45
	18	221.2	38.25	206.7	70.09	214.0 kl	52.84	122.0	16.74	264.8	37.10	193.4 d–j	80.89
	1b	250.3	24.60	415.6	26.73	332.9 b–e	91.51	130.7	18.47	314.2	64.12	222.5 b–e	107.39
	23	419.7	29.45	548.0	45.13	483.8 a	77.09	243.4	20.49	219.9	34.43	231.6 b–d	29.09
	26	279.1	75.43	383.6	62.29	331.3 b–e	84.98	150.4	32.41	183	38.38	166.7 i–m	37.22
	35	330.2	36.01	402.3	15.70	366.3 b	46.36	186.7	25.69	195.6	22.16	191.1 e–j	22.72
	42	225.1	41.08	241.1	31.87	233.1 j–l	35.09	101.9	22.33	79.9	4.82	90.9 p	19.02
	43	190.8	26.63	303.5	20.56	247.1 h–l	64.17	177.4	40.47	207.7	35.99	192.6 d–j	38.97
	55	310.6	30.05	379.5	39.92	345.1 b–d	49.27	147.3	4.55	204.7	21.07	176.0 f–k	33.73
	78b	195.8	33.04	324.3	26.41	260 g–l	74.04	193.8	54.08	120.8	30.89	157.3 j–m	56.46
	83	167.4	44.50	437.7	48.72	302.5 c–h	150.79	199.1	33.07	279.7	38.16	239.4 ab	54.31
	9	186.5	42.29	356.6	57.23	271.5 f–k	102.13	142.2	35.16	204.7	13.83	173.4 g–k	41.58
F3	114	321.7	27.73	253.5	21.01	287.6 d–j	50.27	123.4	17.29	128.7	15.45	126.1 n–p	39.55
	119	309.3	19.76	291.3	34.64	300.3 c–h	27.82	190.5	26.76	208.1	46.86	199.3 c–i	36.56
	12	269.3	66.51	383.4	29.55	326.4 b–f	77.43	188.5	42.90	153.8	33.26	171.2 h–l	40.08
	18	192.9	37.5	214.0	66.17	203.5 l	47.67	132.9	12.89	128.0	25.67	130.5 m–p	54.04
	1b	210.6	29.86	360.2	38.35	285.4 e–j	86.03	151.8	30.76	261.4	61.75	206.6 b–h	73.99
	23	389.3	26.23	500.1	30.32	444.7 a	64.76	197.1	51.67	247.7	15.50	222.4 b–e	44.48
	26	201.2	25.52	349.7	45.73	275.5 e–j	86.46	120.7	6.20	141.9	23.68	131.3 lm–o	19.64
	35	303.1	29.47	284.9	12.58	294.0 d–i	40.94	190.2	19.04	238.4	17.80	214.3 b–f	19.08
	42	178.9	47.00	294.0	1.54	236.5 i–l	68.81	113.1	16.64	75.9	2.94	94.5 op	22.75
	43	147.8	12.36	279.5	33.99	213.6 kl	74.25	152.7	32.65	103.3	0.00	128.0 m–p	33.97
	55	292.6	31.49	421.2	138.04	356.9 bc	115.38	175.9	8.81	243.0	7.91	209.4 b–h	36.70
	78b	327.1	108.05	289.1	28.59	308.1 b–g	75.94	154.7	18.25	110.7	16.95	132.7 l–o	28.62
	83	301.9	113.68	300.4	75.66	301.1 c–h	89.40	193.3	43.89	354.7	0.00	274.0 a	90.95
	9	231.9	1.51	358.0	28.28	294.9 d–h	69.88	109.1	10.60	172.3	70.26	140.7 k–n	57.50
LSD_0.05_						58.341						39.71	

a, b, c, …—In columns, means followed by the same letters are not significantly different.

**Table 3 ijms-24-13905-t003:** Mean values, standard deviations, and homogeneous group for phenolic compounds content (mg g^−1^ of d.w.).

Trait		First Day of Drought (20% of Soil Field Capacity)						After Two Weeks of Drought (Maintaining 20% of Soil Field Capacity)					
Treatment		C		S		Average		C		S		Average	
Genotype		Mean	s.d.	Mean	s.d.	Mean	s.d.	Mean	s.d.	Mean	s.d.	Mean	s.d.
Bingo		19.4	3.57	23.2	1.86	21.3 o	3.38	36.6	2.22	41.8	2.87	39.2 d–g	3.64
F2	114	32.8	2.41	37.9	7.6	35.4 cd	5.9	41.4	6.32	32.3	1.91	36.8 e–j	6.49
	119	27.0	3.46	32.6	3.35	29.8 f–j	4.38	35.9	2.42	30.1	0.64	33.0 i–k	3.50
	12	27.7	3.06	34.3	7.75	30.9 e–h	6.51	27.7	0.86	38.1	5.55	32.9 i–k	6.69
	18	30.1	2.42	24.7	4.50	27.4 g–m	4.42	27.5	3.82	25.3	3.17	26.4 l	3.45
	1b	33.4	0.41	20.9	1.88	27.1 g–m	6.82	38.3	2.27	38.3	4.65	38.3 d–h	3.39
	23	24.79	1.78	25.7	2.02	25.2 k–o	1.82	31.2	2.35	33.5	5.41	32.4 jk	4.04
	26	22.9	0.77	25.5	3.64	24.2 m–o	2.83	31.2	3.63	34.1	1.87	32.6 jk	3.10
	35	26.7	3.36	24.6	1.85	25.7 j–n	2.76	31.6	2.15	31.4	4.10	31.5 k	3.56
	42	33.4	3.57	31.0	0.55	32.2 d–f	2.68	46.3	0.87	42.6	1.84	44.5 a–c	2.40
	43	38.3	0.83	34.6	9.86	36.4 c	6.76	40.6	1.87	39.6	8.26	40.1 c–e	5.57
	55	23.3	1.68	24.8	2.32	24.0 m–o	2.05	36.0	0.28	35.3	5.00	35.6 f–k	3.31
	78b	31.8	4.03	28.1	3.14	29.9 f–i	3.89	33.4	2.43	32.2	8.70	32.8 jk	5.94
	83	28.6	3.09	25.3	1.19	26.9 h–n	2.78	32.5	2.46	40.9	2.63	36.7 e–j	5.05
	9	28.7	0.90	29.4	4.22	29.0 f–l	2.85	34.8	1.97	48.4	2.96	41.6 b–d	7.63
F3	114	43.7	1.07	50.2	6.66	47.0 a	5.07	39.9	4.01	54.2	2.37	47.1 a	6.17
	119	22.1	2.45	30.9	2.23	26.5 i–n	5.20	35.3	3.00	36.4	0.49	35.9 e–k	2.04
	12	27.8	4.93	29.4	2.53	28.6 f–l	3.72	34.7	1.46	39.9	3.81	37.3 d–i	3.85
	18	36.3	3.08	45.6	5.07	41.0 b	5.57	43.8	2.99	51.27	4.07	47.5 a	4.02
	1b	32.0	4.87	26.5	5.51	29.3 f–k	5.64	32.9	1.20	40.0	1.27	36.5 e–j	4.00
	23	19.8	2.06	25.7	3.10	22.74 no	3.98	32.6	2.51	32.9	0.90	32.8 jk	1.80
	26	27.2	1.26	24.0	3.43	25.6 k–n	2.94	28.6	2.81	35.5	2.74	32.0 k	4.46
	35	42.3	3.17	45.9	3.09	44.1 ab	3.04	44.9	2.78	47.8	4.57	46.4 a	4.91
	42	34.2	2.53	28.3	1.41	31.2 e–g	3.68	42.7	4.23	36.7	3.24	39.7 d–f	4.74
	43	37.1	1.39	31.6	1.71	34.3 c–e	3.29	38.2	5.29	31.9	0.00	35.0 g–k	4.84
	55	26.9	0.45	21.9	2.42	24.4 m–o	3.14	30.1	1.92	38.9	3.63	34.7 h–k	5.24
	78b	24.0	2.91	30.1	1.90	27.3 g–m	3.74	31.9	2.96	44.8	3.598	38.4 d–h	7.54
	83	29.3	0.23	25.13	1.89	27.2 g–m	2.53	33.7	4.32	29.7	7.93	31.7 k	6.29
	9	20.5	3.05	29.5	4.48	25.0 l–o	5.99	32.8	0.83	57.9	8.00	45.4 ab	14.4
LSD_0.05_						4.18						4.45	

a, b, c, …—In columns, means followed by the same letters are not significantly different.

**Table 4 ijms-24-13905-t004:** Mean values, standard deviations, and homogeneous group for the mass of stems plant^−1^ (g), the number of grains, and the mass of grains plant^−1^ (g) in control combination (C) and combination with soil drought (D).

Trait		The Mass of Stems/Plant (g)						The Number of Grains						The Mass of Grains/Plant (g)					
Treatment		C		D		Average		C		D		Average		C		D		Average	
Genotype		Mean	s.d.	Mean	s.d.	Mean	s.d.	Mean	s.d.	Mean	s.d.	Mean	s.d.	Mean	s.d.	Mean	s.d.	Mean	s.d.
Bingo		17.2	3.16	4.3	3.09	10.7 c–f	7.30	124.3	17.34	47.8	31.13	86.0 bc	46.40	6.2	0.88	1.8	1.57	4.0 b–d	2.59
F2	114	9.0	4.20	6.2	1.50	7.6 f	3.28	1.0	1.15	2.7	1.70	1.8 i	1.61	0.1	0.05	0.1	0.08	0.1 jk	0.07
	119	14.9	2.86	10.6	1.72	12.7 a–d	3.17	8.5	3.00	7.8	2.87	8.1 i	2.8	0.4	0.07	0.3	0.11	0.4 jk	0.11
	12	16.6	3.52	9.13	1.21	12.9 a–d	4.66	90.8	39.51	76.3	21.39	83.5 b–d	30.42	3.6	1.45	3.0	0.80	3.3 c–f	1.13
	18	13.6	2.22	5.72	1.50	9.7 d–f	4.55	8.3	2.50	0.3	0.50	4.3 j	4.59	0.3	0.09	0.01	0.01	0.2 jk	0.17
	1b	17.3	2.15	13.0	2.87	15.1 a	3.29	76.8	12.69	35.5	21.11	56.1 d–f	27.32	3.5	0.43	1.5	0.86	2.5 e–h	1.23
	23	15.0	2.45	11.3	0.77	13.2 a–c	2.56	58.8	17.21	37.0	4.69	47.9 e–g	16.48	2.4	0.75	1.5	0.24	1.9 g–i	0.73
	26	13.7	3.18	9.2	1.20	11.5 c–e	3.28	44.5	17.18	43.3	10.63	43.9 e–h	13.24	2.0	0.69	2.0	0.45	1.9 g–i	0.54
	35	14.8	1.18	7.8	1.51	11.3 c–e	3.94	69.3	23.91	58.3	10.40	63.7 c–e	18.05	3.0	0.80	2.6	0.39	2.8 d–g	0.62
	42	14.1	1.96	7.9	2.90	11.0 c–e	4.01	16.0	14.21	25.8	15.13	20.9 h–j	14.55	0.9	0.80	1.3	0.87	1.1 i–k	0.83
	43	12.9	1.66	9.4	1.99	11.1 c–e	2.55	29.8	14.45	17.8	21.6	23.8 g–j	18.19	1.4	0.65	0.9	1.19	1.1 i–k	0.92
	55	14.2	1.97	9.7	1.07	11.9 a–e	2.82	26.3	7.41	51.5	17.62	38.9 e–i	18.40	1.2	0.29	2.2	0.76	1.7 g–i	0.75
	78b	16.8	3.28	8.9	1.05	12.8 a–d	4.79	132.0	73.14	82.0	9.27	107.0 ab	55.17	5.1	2.40	3.3	0.39	4.2 bc	1.84
	83	12.7	1.02	6.9	1.25	9.8 d–e	3.28	139.5	36.57	104.0	22.38	121.8 a	33.88	5.4	1.07	4.0	0.76	4.7 ab	1.15
	9	12.9	2.53	9.2	5.21	11.0 c–e	4.27	141.8	21.61	104.0	70.15	122.9 a	52.12	7.0	2.73	4.0	2.71	5.5 a	3.00
F3	114	11.3	5.01	9.8	2.45	10.6 c–f	5.08	2.5	1.07	0.5	0.09	1.5 j	0.55	0.1	0.01	0.03	0.02	0.07 k	0.04
	119	18.9	6.16	11.1	1.25	15.0 a	5.84	9.3	3.20	9.8	1.50	9.5 ij	2.33	0.4	0.09	0.4	0.04	0.4 jk	0.07
	12	13.1	2.38	9.1	0.34	11.1 c–e	2.68	103.3	25.2	74.8	15.22	89.0 bc	24.58	4.0	1.09	3.0	0.53	3.5 b–e	0.97
	18	7.8	3.07	7.6	2.99	7.7 f	4.92	9.3	2.94	3.3	0.74	6.3 j	2.73	0.07	0.02	0.02	0.01	0.05 k	0.05
	1b	17.3	4.16	12.4	0.82	14.8 ab	3.80	62.8	31.6	47.5	13.4	55.1 d–f	23.9	2.7	1.31	2.0	0.63	2.4 e–h	1.02
	23	18.0	1.88	9.6	1.60	13.8 a–c	4.80	76.3	18.1	27.5	8.35	51.9 e–g	29.1	3.0	0.64	1.1	0.32	2.1 f–i	1.16
	26	13.4	1.55	9.4	0.91	11.4 c–e	2.45	49.3	14.5	39.88	5.56	44.5 e–h	11.4	2.4	0.75	1.8	0.29	2.1 f–i	0.62
	35	11.3	4.07	7.2	2.91	9.3 ef	4.58	97.0	20.7	78.7	27.27	87.9 bc	22.61	2.9	0.17	2.2	0.27	2.6 e–g	0.97
	42	11.8	1.97	8.0	6.70	9.9 d–f	5.00	18.5	13.58	27.8	32.95	23.1 g–j	23.85	0.7	0.52	1.3	1.69	1.0 i–k	1.20
	43	15.1	2.14	8.4	0.238	11.7 b–e	3.82	43.0	42.58	48.0	16.17	45.5 e–h	29.93	1.9	1.83	2.3	0.48	2.1 f–i	1.26
	55	13.9	2.33	7.5	0.970	10.7 c–f	3.79	54.3	24.30	57.0	11.34	55.6 d–f	17.61	2.5	1.01	2.3	0.65	2.4 e–h	0.80
	78b	19.8	2.44	7.4	0.69	13.6 a–c	6.85	167.5	56.97	84.8	18.39	126.1 a	59.10	6.2	1.17	3.4	0.62	4.7 ab	1.76
	83	6.3	6.31	2.1	2.93	4.2 g	5.06	48.0	55.43	11.5	22.34	29.8 f–j	43.72	2.2	2.48	0.4	0.81	1.3 h–j	1.94
	9	13.6	1.70	8.4	1.91	11.0 c–e	3.27	119.0	26.14	91.3	23.41	105.1 ab	27.35	4.7	0.88	3.4	1.10	4.1 bc	1.15
LSD_0.05_						3.2						29.80						1.2	

a, b, c, …—In columns, means followed by the same letters are not significantly different.

**Table 5 ijms-24-13905-t005:** The effect of excised-leaf water loss (ELWL) values, independent: after 0–3 h, after 4–6 h, and after 0–6 h (independent variable, *x*), on the mass of stems plant^−1^, the number of grains, the mass of grains plant^−1^ (dependent variable, *y*) estimated by regression model *y* = a *x* + b.

Independent Variable	Dependent Variable	Model	Percentage Variance Accounted	Standard Error of Observations
ELWL after 0–3 h	The mass of stems plant^−1^	*y* = 21.53 *** − 0.314 * *x*	3.0	4.87
	The number of grains	*y* = 20.184 *** − 0.028 * *x*	3.2	4.86
	The mass of grains plant^−1^	*y* = 20.109 *** − 0.637 * *x*	2.6	4.88
ELWL after 4–6 h	The mass of stems plant^−1^	*y* = 15.14 *** − 0.285 *x*	2.0	5.23
	The number of grains	*y* = 13.051 *** − 0.0057 *x*	-	5.3
	The mass of grains plant^−1^	*y* = 12.935 *** − 0.074 *x*	-	5.3
ELWL after 0–6 h	The mass of stems plant^−1^	*y* = 33.23 *** − 0.500 * *x*	3.0	7.77
	The number of grains	*y* = 30.39 *** − 0.0286 *x*	0.8	7.86
	The mass of grains plant^−1^	*y* = 30.23 *** − 0.611 *x*	0.4	7.88

* *p* < 0.05; *** *p* < 0.001.

## Data Availability

The data in this manuscript are available from the corresponding author upon reasonable request.

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
