# Peer review of "Effect of Soil Drought Stress on Selected Biochemical Parameters and Yield of Oat × Maize Addition (OMA) Lines"

_ijms, 2023, doi:10.3390/ijms241813905_

Round 1
Reviewer 1 Report
The manuscript aims to identify OMA lines generated through wide crossing - using maize as pollinator - to investigate possible differences based on leaf water loss and other selected biochemical and agronomic changes under simulated drought conditions.
The article is well written and presented. The introduction provides enough background to understand the topic. The methods are adequately described and all conclusions are supported by the results. The manuscript requires some changes in order to be published. I have some minor revisions described below.
I feel that the abstract could highlight the aims of the study and be more descriptive about the results. A big portion of it is introduction.
In some parts of the text the authors named plants as "objects" eg. lines 197, 201, 209 and so on. Why not call it plant? I suggest to change for a more appropriate term.
I would place the PCs values in a better place on the figure 4 or put it also on the caption of the figure. Same for fig. 5.
Table 5. I think "the mass of stem..." could be replaced for "the mass..." only.
Line 391: Please, use a citation that relates also to the field of study, in this case, drought.
Line 403: Please insert a citation number on "Sinay and Karuwal 403 (2014)".
The reader would have a better understanding of the impact of your study if the discussion's last paragraph had a conclusion with impacts and future perspectives of your manuscript.
Author Response
Response to Reviewer 1 Comments
Reviewer #1
Point 1: The manuscript aims to identify OMA lines generated through wide crossing - using maize as pollinator - to investigate possible differences based on leaf water loss and other selected biochemical and agronomic changes under simulated drought conditions.
The article is well written and presented. The introduction provides enough background to understand the topic. The methods are adequately described and all conclusions are supported by the results. The manuscript requires some changes in order to be published. I have some minor revisions described below.
Response: Thank you very much.
Point 2: I feel that the abstract could highlight the aims of the study and be more descriptive about the results. A big portion of it is introduction.
Response:
Thank you very much for the precious comment. We incorporated the requested changed into the abstract, we add information about the aims of the study and put more details about the general results of the study. Therefore new version of the abstract is more informative for the readers even without going through the entire manuscript.
Point 3: In some parts of the text the authors named plants as "objects" eg. lines 197, 201, 209 and so on. Why not call it plant? I suggest to change for a more appropriate term.
Response:
We reconsidered the nomenclature use in the manuscript, and we agree with the Reviewer that using just the name plants is more relevant since our objects of the experiment are plants therefore we change the term in line 197, 201, 209 and we check it throughout the entire manuscript and change the term “objects” to “plants”.
Point 4: I would place the PCs values in a better place on the figure 4 or put it also on the caption of the figure. Same for fig. 5.
Response: The percentage of explanation is placed as far off-axis as possible so that the Figure is as visible as possible. We have also added these values in the captions of the Figures.
Point 5: Table 5. I think "the mass of stem..." could be replaced for "the mass..." only.
Response:
Thank you for the suggestion, but we use the term “the mass of stem” to be more specific not to make an impression that we consider the whole mass of the plant (e.g. stem and roots). We analyzed only above the ground mass of the plants actually only stem to be more specific, therefore I hope the explanation would be accepted by the Reviewer and the Editor.
Point 6: Line 391: Please, use a citation that relates also to the field of study, in this case, drought.
Response:
We have added two appropriate references related to drought, these means: Warzecha, T.; Bathelt, R.; Skrzypek, E.; WarchoÅ‚, M.; Bocianowski, J.; Sutkowska, A. Studies of Oat-Maize Hybrids Tolerance to 687 Soil Drought Stress. Agriculture 2023, 13, 243. https://doi.org/10.3390/agriculture13020243, and Oleksiak, T.; Spyroglou, I.; PacoÅ„, D.; Matysik, P.; Pernisová, M.; Rybka, K. Effect of drought on wheat production in Poland 723 between 1961 and 2019. Crop Sci. 2021, 62, 728–743.
Point 7: Line 403: Please insert a citation number on "Sinay and Karuwal 403 (2014)".
Response: We have corrected it. Thank you very much for pointing out our mistake.
Point 8: The reader would have a better understanding of the impact of your study if the discussion's last paragraph had a conclusion with impacts and future perspectives of your manuscript.
Response:
We have added the last paragraph in the Discussion section which serves the impact and perspectives of our research in applied aspect. It is as follow: “These associations might find possible application in crop improvement programs since the excised leaves water loss (ELWL test) as well as amount of phenolics and sugars significantly correlated with yield of plants under soil drought. These physiological and biochemical features can be recommended in plants breeding as a fast screening test of the germplasm possessing higher soil drought tolerance”.

Reviewer 2 Report
Dear Authors,
In this manuscript, the authors investigated the effect of soil drought stress on selected biochemical parameters and yield of Oat × Maize addition lines. The topic is of interest but there are several issues that should be addressed before I can indicate this MS for publication.
There are several grammatical issues, including punctuation (many commas are missing, for example, in several sentences) and the sentence are too big. Please carefully read the MS and split some of the sentences in more than one, using the proper punctuation, so that the reading can be easier to the reader.
Also please avoid calling control objects to control plants, as plants are not objects.
I think that the MS has potential to published, but first the authors should properly revise all the document.
Specific comments:
Abstract
The abstract is to theoretical, it would be better to give more focus to the aims and results.
Introduction
In general, this section is to extensive. I advise the authors to reduce some parts and to be more straightforward. Also, there are a lot of references missing. Most of the sentences is not supported by any reference. For instance, a great attention is given to photosynthesis, but then the authors do not present any result related to this physiological process. Please focus the introduction on presenting only literature important to study and relate your results with your research goals.
Other minor comments:
Line 60: please change it to: “plants that are more tolerant to drought stress”.
Line 98-99: “A large group of 98 complex phenols are tannins, also called tannins, and flavonoids”. Please rewrite.
Line 101: “The toxic properties of phenols include: on protein denaturation.” Please rewrite.
Results:
Table 1 and text: the number of decimal places can and should be reduced.
Figure 2 and 3: the meaning of the asterisk should be given in the figures caption,
Figure 4 and 5: why did the authors did the split between the control and drought stressed plants? Wouldn’t be helpful to put all of them together and see if the PCA allows the separation between control and drought stress plants? Also, the figures showing the loading factor are missing, and should be included. The % of explanation should also be given outside the axis. Please see other published articles to see how this kind of graphs is usually presented.
Discussion:
Line 374-377: Please rewrite de sentence, there are grammatical issues, and the sentence is too big.
Line 388-391: The authors do not present any kind of results related to photosynthesis. So I do not understand the relevance of this sentence to the discussion.
Line 443: compared to control combination with sufficient access to water. What do you mean by “control combination with sufficient access to water? If is a control plant it should have sufficient access to water, right?
Material and Methods
Section 4.3: Please describe the greenhouse conditions during the experimental period. Also an explanation how the field capacity was determined and monitored should be added.
Section 4.5: please refer the standard that was used to calculate the total amount of sugars. If other method was used to calculate this amount, please describe the equation used for such calculations.
There are several grammatical issues, including punctuation (many commas are missing, for example, in several sentences) and the sentence are too big. I advised the authors to split some of the sentences in more than one, using the proper punctuation, so that the reading can be easier to the reader.
Author Response
Response to Reviewer 2 Comments
Reviewer #2
Point 1: In this manuscript, the authors investigated the effect of soil drought stress on selected biochemical parameters and yield of Oat × Maize addition lines. The topic is of interest but there are several issues that should be addressed before I can indicate this MS for publication.
Response: Thank you very much.
Point 2: There are several grammatical issues, including punctuation (many commas are missing, for example, in several sentences) and the sentence are too big. Please carefully read the MS and split some of the sentences in more than one, using the proper punctuation, so that the reading can be easier to the reader.
Response: We have corrected the manuscript grammatically.
Point 3: Also please avoid calling control objects to control plants, as plants are not objects.
Response:
We reconsidered the nomenclature use in the manuscript, and we agree with the Reviewer that using just the name plants is more relevant since our objects of the experiment are plants therefore we change the term and we check it throughout the entire manuscript and change the term “objects” to “plants”.
Point 4: I think that the MS has potential to published, but first the authors should properly revise all the document.
Response:
We revised the whole MS as Reviewer requested.
Specific comments:
Abstract
Point 5: The abstract is to theoretical, it would be better to give more focus to the aims and results.
Response:
Thank you very much for the comment. We incorporated the requested changed into the abstract, we add information about the aims of the study and put more details about the general results of the study.
Point 6: Introduction
In general, this section is to extensive. I advise the authors to reduce some parts and to be more straightforward. Also, there are a lot of references missing. Most of the sentences is not supported by any reference. For instance, a great attention is given to photosynthesis, but then the authors do not present any result related to this physiological process. Please focus the introduction on presenting only literature important to study and relate your results with your research goals.
Response:
We have corrected the Introduction. We reduced numerous unnecessary information regarding photosynthesis, proteins and biotic stresses. We also eliminated seven references from this part of text.
Other minor comments:
Point 7: Line 60: please change it to: “plants that are more tolerant to drought stress”.
Response: We have corrected this sentence.
Point 8: Line 98-99: “A large group of 98 complex phenols are tannins, also called tannins, and flavonoids”. Please rewrite.
Response: We have rewrite the sentence.
Point 9: Line 101: “The toxic properties of phenols include: on protein denaturation.” Please rewrite.
Response: We have rewrite the sentence.
Results:
Point 10: Table 1 and text: the number of decimal places can and should be reduced.
Response: We have reduced the number of decimal places.
Point 11: Figure 2 and 3: the meaning of the asterisk should be given in the figures caption,
Response: We have added explanations of the meaning of the asterisk in the captions of Figures 2 and 3.
Point 12: Figure 4 and 5: why did the authors did the split between the control and drought stressed plants? Wouldn’t be helpful to put all of them together and see if the PCA allows the separation between control and drought stress plants? Also, the figures showing the loading factor are missing, and should be included. The % of explanation should also be given outside the axis. Please see other published articles to see how this kind of graphs is usually presented.
Response: The results for PCA were separated into control and drought treatments because for the control plants we performed ELWL test, which was initial for checking the rate of water loss in certain time (the process is regulated by various internal factors – anatomical and physiological dependent on the genotype). Doing it separately enable us to detect if the genotypes are stable in control and drought conditions. If we merge control and drought data, we will lose some data. We supplemented the description of the results with the numerical values of the loading factors. The percentage of explanation is placed as far off-axis as possible so that the Figure is as visible as possible.
Discussion:
Point 13: Line 374-377: Please rewrite de sentence, there are grammatical issues, and the sentence is too big.
Response: We have rewrite the sentence as follow: “Different crops, and within them different genotypes, having low values of ELWL have a greater ability to maintain water balance in the leaves, largely due to soluble sugars in the plant [45,46]”.
Point 14: Line 388-391: The authors do not present any kind of results related to photosynthesis. So I do not understand the relevance of this sentence to the discussion.
Response: We agree with the comment and we delete these information.
Point 15: Line 443: compared to control combination with sufficient access to water. What do you mean by “control combination with sufficient access to water? If is a control plant it should have sufficient access to water, right?
Response: We agree with the Reviewer. That is right that control plants have sufficient access to water, therefore we corrected the sentence as follow: “Studies conducted on wheat showed lower biomass values in plants growing in drought conditions compared to control ones”.
Material and Methods
Point 16: Section 4.3: Please describe the greenhouse conditions during the experimental period. Also an explanation how the field capacity was determined and monitored should be added.
Response: We added the information about weather conditions and the method of checking the field water capacity.
Point 17: Section 4.5: please refer the standard that was used to calculate the total amount of sugars. If other method was used to calculate this amount, please describe the equation used for such calculations.
Response: We supplemented the missing information: “The content of sugars was calculated in mg of sucrose per 1 g of dry matter of plant tissue [mg g–1 DM]”.

Reviewer 3 Report
The paper provides little new information and is very specific, so it's out of the scope of a generalistic journal such as IJMS.
The main problem is that the experiments were performed without the appropriate controls. The authors constructed hybrid lines, but they did not compare their drought tolerance with the two parental lines. Thus, the main open question is: Is constructing these hybrids useful for generating drought-tolerant crops? Is the yield better than that of the parental lines? According to the 'Materials and Methods' section, the cultivars used were:
"Maize variety Waza (positive control) and oat variety Stoper (negative control)."
However, in the experiments, only the Bingo cultivar appears, and it is not explained why this cultivar was used. As a result, it appears that parental lines are not included in the report. Therefore, one cannot properly judge the agronomic effects of the hybridization.
Additionally, the amount of molecular information is very limited, and the main conclusion seems to be:
"The content of phenolic compounds might be used as a biochemical indicator of plant drought tolerance since there was a significant correlation with high kernel yields of plants subjected to drought stress."
However, this is something that has been known for a long time. For instance, refer to:
https://www.mdpi.com/1420-3049/24/13/2452
Another point is that repeated numbers appear in the columns:
For instance, in table 3, the numbers for line 18 are identical in F2 and F3.
In table 4, the values for lines 114, 18, and 35 are again identical between F2 and F3.
Statistically, obtaining exactly the same numbers in different experiments with different plants is almost impossible. Do the authors have any explanation for this mathematical anomaly, or is it a mistake in the data processing?
English is fine
Author Response
Response to Reviewer 3 Comments
Reviewer #3
Point 1: The paper provides little new information and is very specific, so it's out of the scope of a generalistic journal such as IJMS.
Response: The material used for this study is unique and generated by our team. To our knowledge, this is the first time when OMA lines response to soil drought stress was expressed in biochemical and physiological changes associated with yield, what is the applied aspect of research. To detect OMA lines we used retrotransposon sequences as a molecular aspect of the work. Therefore in our opinion the investigation which we done is in the scope of IJMS generally, and especially to the special issue devoted to biochemical changes generated by drought stress. The topic of the Special Issue is “New Strategies for Drought Tolerance of Crops: Physiological, Biochemical, and Molecular Aspects.”
Point 2: The main problem is that the experiments were performed without the appropriate controls. The authors constructed hybrid lines, but they did not compare their drought tolerance with the two parental lines. Thus, the main open question is: Is constructing these hybrids useful for generating drought-tolerant crops? Is the yield better than that of the parental lines? According to the 'Materials and Methods' section, the cultivars used were:
"Maize variety Waza (positive control) and oat variety Stoper (negative control)."
Response: The hybrids were generated not as it is commonly done to create mapping population, it means not by crossing two parents and generate full progeny possessing the whole sets of parental genomes. The OMA lines are not regular hybrids. It is a kind of side effect of haploidization process, while producing haploids and doubled haploids (DH) of oat with the application of wide crossing with maize. It is impossible to obtain high number of OMA lines from one crossing, sometimes there is e.g. one OMA line from one crossing, and rest of plants are regular haploids. Even if we obtained couple of OMA lines, they are not always fertile – do not produces grains. To obtain next generation for research we need to possess appropriate amount of grains.
We also would like to add that even in literature in previous time they were called partial hybrids, since they possess only a part of set of maize chromosomes or even some fragments of maize chromatin.
We would like to also explain the meaning of positive and negative control. In molecular part of research we used maize cv. Waza, which possess many copies of retrotransposon Grande-1 on each chromosome. So, if we detected fragment of 500 bp retrotransposon Grande-1 on PCR reaction, it means that the reaction was properly performed. In other hand there should not be 500 bp product in PCR with oat DNA application (this means oat cv. Stoper). If the plants obtained by wide crossing had 500 bp product, it means that the part of maize genome was incorporated to the oat genome. That positive and negative controls were for molecular identification of OMA lines to distinguish them from DH lines, as maize possess Grande-1 and oat does not. Control for OMA lines in drought experiment is described in paragraph 4.3 – “Greenhouse experiment”. Here we only mention in a short wat that cv. ‘Bingo’ was use in the experiment with drought because it was in the pedigree of oat lines used for wide crossing with maize. In this experiment control was not only cv. ‘Bingo’, but also all OMA lines under optimal water regime.
Point 3: However, in the experiments, only the Bingo cultivar appears, and it is not explained why this cultivar was used. As a result, it appears that parental lines are not included in the report. Therefore, one cannot properly judge the agronomic effects of the hybridization.
Response: The cv. ‘Bingo’ was use in the experiment with drought because it is still (more than 30 years) one of the best oat cultivar which was in the pedigree of oat lines used for wide crossing with maize. Actually there were no parental lines as it was a crosses between oat lines and maize cv. Waza as pollinator, not crosses of oat with oat. We described that in the paragraph 4.1. “Molecular identification of OMA lines”: Intraspecific oat hybrids (cross combinations of cultivars or breeding lines) pollinated with maize allowed to generate a population of DH lines and OMA lines equivalent to the classic F2 generation, but with the difference that only homozygous forms constituted this population”. We evidenced that the highest number of maize chromosomes added to the oat genome was four chromosomes (PeerJ 2018, 6, e5107). This reference is cited in Introduction and we added it to the M&M.
We would not like to judge the agronomic effects of the hybridization, as it was not full hybridization, but only partial hybridization. In the literature there is no report talking about retaining whole set of maize chromosomes into oat genome. We would like to judge if introgression of part of the maize genome have impact on drought stress tolerance expressed in measured traits.
Point 4: Additionally, the amount of molecular information is very limited, and the main conclusion seems to be:
"The content of phenolic compounds might be used as a biochemical indicator of plant drought tolerance since there was a significant correlation with high kernel yields of plants subjected to drought stress."
However, this is something that has been known for a long time. For instance, refer to:
https://www.mdpi.com/1420-3049/24/13/2452
Response: We agree with the reviewer that the content of phenolic compounds is very often described as a biochemical indicator of plant tolerance to drought, but we want to emphasize that for the first time this relationship has been correlated with the yield of oat-maize hybrids. We would like to stress that the material created by us and used for research is unique. The conclusion presented by Reviewer is reduced only to one mentioned above. In the manuscript we described more results summarized by conclusions and applied advices. At first, we applied molecular method to screen plants in order to detect maize genome introgression, since producing oat DH lines the retaining of maize chromosomes is not checked, assuming that the maize chromosomes (as alien) are eliminated during subsequent cells divisions. Also, beside phenolics we checked sugars and water content in tissues (ELWL test), what gives opportunities for fast screening of plants. Beside the mentioned above conclusions we checked if the reaction was stable in subsequent generations (F2 and F3). We also indicated OMA lines no. 9 and 78b as the best candidates for further investigation and possible application in breeding program on oat resistance to soil drought stress.
Point 5: Another point is that repeated numbers appear in the columns:
For instance, in table 3, the numbers for line 18 are identical in F2 and F3.
Response:
We would like to thank the Reviewer for the indication of the mistake with the repeated number in table 3. We checked again all the data and we found out that it is just simple mistake while coping the data. By mistake while coping we put into the table twice information for F2 population, so we correct it and we put proper values for F3 generations for genotype 18, 35 and 114. All the changes are visible thanks Track change mode.
Point 6: In table 4, the values for lines 114, 18, and 35 are again identical between F2 and F3. Statistically, obtaining exactly the same numbers in different experiments with different plants is almost impossible. Do the authors have any explanation for this mathematical anomaly, or is it a mistake in the data processing?
Response:
We completely agree with the Reviewer that obtaining exactly the same numbers in different experiments with different plants is almost impossible. The problem was similar like explained in point 5. We checked again all the data and correct the results for F3 generation for genotype 18,35 and 114. All the changes are visible thanks Track change mode.

Round 2
Reviewer 3 Report
I have carefully read the author's response. I appreciate the detailed answer. I also appreciate the effort made by the authors to develope a novel technique to generate new OMA lines with increased drought resistance. But there are some essential question that remains unanswered.
According to the authors:
Actually there were no parental lines as it was a crosses between oat lines and maize cv. Waza as pollinator, not crosses of oat with oat.
So. Which oat lines were used in the crosses? Why they are not described nor included in the experiments as negative control? Authors refer that Cv Bingo is very standard on Oat, but, this cultivar is the one that has been pollinized with maize? Without this control is difficult to judge whether this system really creates drought resistant lines.
Another question is: How are the OMA lines from the agronomical point of view? Are they similar to oat (size, shape, phenology)? is the cultivation similar.? I would like authors to include pictures of the greenhouse experiments where OMA lines are compared to the control oat with normal irrigation and under drought stress.
Author Response
Response to Reviewer 3 Comments
Reviewer #3
Point 1: I have carefully read the author's response. I appreciate the detailed answer. I also appreciate the effort made by the authors to develope a novel technique to generate new OMA lines with increased drought resistance. But there are some essential question that remains unanswered.
Response: Thank you very much, we would like to state that all the critics expressed by the Reviewer #3 we took into consideration as the option to improve the manuscript. We hope that our further explanation presented in Point 2 and Point 3 and the changes made thanks the Reviewer will help to conform the manuscript to required form.
Point 2: So. Which oat lines were used in the crosses? Why they are not described nor included in the experiments as negative control? Authors refer that Cv Bingo is very standard on Oat, but, this cultivar is the one that has been pollinized with maize? Without this control is difficult to judge whether this system really creates drought resistant lines.
Response: We would like to recall the section from chapter Material and methods: “The plant material for the study were obtained by wide crossing of oat with maize, at The Franciszek Górski Institute of Plant Physiology, Polish Academy of Sciences in Krakow. Intraspecific oat hybrids (cross combinations of cultivars or breeding lines) pollinated with maize allowed to generate a population of DH lines and OMA lines equivalent to the classic F2 generation, but with the difference that only homozygous forms constituted this population [36,50]. A total of 120 descendants of wide crosses were tested to detect oat-maize hybrids.”
In order to create a new cultivars, two parents are crossed with each other and a heterozygous population of plants is obtained, which is brought to homozygosity over several years. One of the methods of accelerating breeding is to derive homozygous lines, e.g. by distant crossings. As a result of crossing oat with maize, we obtain not only homozygous lines (doubled haploids), but also as a result of incomplete elimination of the maize genome, an oat-maize hybrid. We would like to explain more precisely that F1 intraspecific hybrids of oat were the cross combination of breeding lines or cultivars and the F1 plants were pollinated with maize to obtain haploids and then doubled haploids, as side effect there were also OMA plants appeared, but very rarely. Actually the plants we obtain were the equivalent of F2 generation. So the plants obtained were a part of the F2 generation and as a result of recombination resulted in a whole range of genotypes that were very different from the parental forms in consequence of random genetic recombination (segregation of chromosomes in the first meiotic division and the crossing-over phenomenon), while the OMA forms with retained chromosomes maize were often single individuals. The progeny lines (OMA and DH) taking into account that sometimes it was single OMA line from one crossing combination it could be implicated that the descendants were more different from parental form than OMA line from cv. Bingo in terms of the expression of oat genome features and the most differentiating factor were additional chromosomes of maize. Therefore, the parental line control of oat would not bring anything more to the assessment in relation to the comparison of OMA lines with cv. Bingo.
We were not aimed to assess the heterosis effect of the hybrid in relation to the parents, or the effect of transgression in the offspring generation, but to assess the impact of retained maize chromosomes on selected oat traits (in OMA lines).
Therefore, the control consisted of oat parental lines would not bring anything more to the assessment in relation to what the comparison of lines OMA with Bingo variety. Moreover we would like to explain that parental forms were F1 generation - (not oat lines) it means cross combination of different oat lines - pollinated with maize.
In our research, we focused on comparing the OMA lines with retained chromosomes of maize with the typical pure oat genome represented by the cv. Bingo. Maize possess C4 photosynthesis pattern and oat C3. In some reports by Kynast et al. 2002 and Kowles et al. 2008 it was postulated that retained maize chromosomes might transfer PEPC (phosphoenolpyruvate carboxylase), enzyme typical for maize (generally C4 plants) activity into OMA plants but the level never reach the one observed in maize. Therefor we choose the model of OMA plants generated by our team and compare with typical oat genome denoted by cv. Bingo. We check other traits which might be associated with resistance to drought stress and yield formation under control and soil drought condition in plants possess extra maize chromosomes represented by a set of OMA lines.
At the end we would like to add that we found in references connected with the topic e.g. articles by Kynast et al. 2002 and Rines 2009 where the reference pure genome was an oat cultivar. Also in the article by Skrzypek at al. Complex characterization of oat (Avena sativa L.) lines obtained by wide crossing with maize (Zea mays L.). PeerJ 2018, 6, e5107 the reference oat genome was a cultivar.
We would like to stress that our research is the first report according to stability of OMA lines under soil drought stress in subsequent generation F2 and F3. But we would like thank to the Reviewer for the inquisitiveness because it helped us to put simple correction of the section from chapter Material and methods as follows:
“The plant material for the study were obtained by wide crossing of oat with maize, at The Franciszek Górski Institute of Plant Physiology, Polish Academy of Sciences in Krakow. Intraspecific oat F1 hybrids (cross combinations of cultivars or breeding lines) pollinated with maize allowed to generate a population of DH lines and OMA lines equivalent to the classic F2 generation, but with the difference that only homozygous forms constituted this population [36,50]. A total of 120 descendants of wide crosses were tested to detect oat-maize hybrids.”
Point 3: Another question is: How are the OMA lines from the agronomical point of view? Are they similar to oat (size, shape, phenology)? is the cultivation similar.? I would like authors to include pictures of the greenhouse experiments where OMA lines are compared to the control oat with normal irrigation and under drought stress.
Response: Oat-maize hybrids which we used in this experiment were oat-shaped. However we expected that some of the maize genes added to the oat genome might be expressed. Oat is a plant that performs C3 photosynthesis, which type is associated with a significant occurrence of photorespiration, which under current atmospheric conditions reduces the potential of plants by 40% (Matsuoka et al. 2001). Under stress conditions, these values increase. In plants carrying out C4 photosynthesis, photorespiration is severely limited by increasing the concentration of carbon dioxide in the bundle sheaths. The advantage of C4 plants over C3 is revealed especially in high temperature conditions. In addition, thanks to the mechanism of carbon dioxide concentration, rapid assimilation is possible also in drought conditions due to the lower demand for water. Maize is a C4 photosynthesis plant. It is characterized by a transpiration coefficient of 350, while in oat this value is over 600. It has been proven that the OMA lines, which have additional maize chromosomes, exhibit the activity of the enzymes phosphoenolpyruvate carboxylase PEPC and orthophosphate pyruvate dikinase PPDK, which are found only in plants carrying out type C4 photosynthesis (Kowles et al. 2008).
OMA lines have not been included in breeding programs so far, so they are not in cultivation. However, research on the characteristics of the OMA lines and their tolerance to various types of environmental stresses is still ongoing and creates opportunities to expand the genetic pool of oats used to breed new cultivars. Until it is certain that the OMA lines are stable hybrids, it will not be possible to introduce them into breeding programs.
I would also like to thank to the Reviewer for the idea of incorporating photos from our experiment. It would certainly improve the manuscript. Since we reported the experiment also photographically, we would like to follow the Reviewer suggestion and prepared Figure 6 and put it into the revised (Revision 2) version of the manuscript.

Round 3
Reviewer 3 Report
This version has been substantially improved and is more easy to understand.
I will recommend publication.